# Three and Five-Year Mortality in Ovarian Cancer after Minimally Invasive Compared to Open Surgery: A Systematic Review and Meta-Analysis

**DOI:** 10.3390/jcm9082507

**Published:** 2020-08-04

**Authors:** Floriane Jochum, Muriel Vermel, Emilie Faller, Thomas Boisrame, Lise Lecointre, Cherif Akladios

**Affiliations:** Department of Gynecology, Strasbourg University Hospital, 67000 Strasbourg, France; floriane.jochum@chru-strasbourg.fr (F.J.); muriel.vermel@chru-strasbourg.fr (M.V.); emilie.faller@chru-strasbourg.fr (E.F.); thomas.boisrame@chru-strasbourg.fr (T.B.); lise.lecointre@chru-strasbourg.fr (L.L.)

**Keywords:** ovarian cancer, laparoscopy, minimally invasive surgery, survival, mortality

## Abstract

As regards ovarian cancer, the use of minimally invasive surgery has steadily increased over the years. Reluctance persists, however, about its oncological outcomes. The main objective of this meta-analysis was to compare the three and five-year mortality of patients operated by minimally invasive surgery (MIS) for ovarian cancer to those operated by conventional open surgery (OPS), as well as their respective perioperative outcomes. PubMed, Cochrane library and CinicalTrials.gov were systematically searched, using the terms laparoscopy, laparoscopic or minimally invasive in combination with ovarian cancer or ovarian carcinoma. We finally included 19 observational studies with a total of 7213 patients. We found no statistically significant difference for five-year (relative risk (RR) = 0.89, 95% CI 0.53–1.49, *p* = 0.62)) and three-year mortality (RR = 0.95, 95% CI 0.80–1.12, *p* = 0.52) between the patients undergoing MIS and those operated by OPS. When five and three-year recurrences were analyzed, no statistically significant differences were also observed. Analysis in early and advanced stages subgroups showed no significant difference for survival outcomes, suggesting oncological safety of MIS in all stages. Whether the surgery was primary or interval debulking surgery in advanced ovarian cancer, did not influence the comparative results on mortality or recurrence. Although the available studies are retrospective, and mostly carry a high risk for bias and confounding, an overwhelming consistency of the evidence suggests the likely effectiveness of MIS in selected cases of ovarian cancer, even in advanced stages. To validate the use of MIS, the development of future randomized interventional studies should be a priority.

## 1. Introduction

Ovarian cancer is the most lethal of common tumours in women, and represents the fifth most frequently diagnosed neoplasm among women [1]. Surgery, together with chemotherapy, are the pillars of the management of ovarian cancer. For early stages, the main objective is to establish the stage of the disease with the purpose of confirming the indication of adjuvant chemotherapy. For advanced tumors, the mainstay of the curative treatment is radical cytoreduction without any residual disease, followed by chemotherapy [2]. Whenever this finality is unachievable with upfront surgery, neoadjuvant chemotherapy and interval debulking surgery were accepted as valid alternatives.

The standard surgical approach is open surgery (OPS). In selected cases, minimally invasive surgery (MIS) has been shown to be safe in terms of postoperative complications and short term mortality [3,4]. Reluctance persists, however, about its oncological outcomes in the longer term. Over the last ten years, multiple studies have compared survival in ovarian cancer after MIS and OPS without showing any clear differences between the approaches, but each of these studies had a limited sample size. Meta-analyses of ovarian cancer have recently been published [5,6,7,8,9,10,11,12] showing similar operative and clinical outcomes between patients treated by MIS and those operated by OPS. However, none of them compared three or five-year survival or performed an overall analysis (early and advanced stages). There is a need to evaluate and pool the relevant data together in a systematic review and meta-analysis to provide more robust evidence regarding survival after MIS versus OPS.

The main objective of this meta-analysis was to compare the survival of patients operated by laparoscopy for ovarian cancer to those operated by conventional open surgery, as well as their respective perioperative outcomes.

## 2. Experimental Section

This was a systematic review and meta-analysis which followed a detailed a priori study protocol. It has been registered in PROSPERO (CRD42020183284). It was conducted according to the Meta-analysis of Observational Studies in Epidemiology guidelines [13] and Preferred Reporting Items for Systematic Reviews and Meta-Analyses (PRISMA) guidelines [14] (Appendix A).

PubMed, Cochrane library and CinicalTrials.gov were systematically searched, using the terms laparoscopy, laparoscopic or minimally invasive in combination with ovarian cancer or ovarian carcinoma. The literature search was limited to articles published in the last ten years (from January 2009 to April 2020).

Using these search criteria, we identified all English language original studies comparing outcomes of patients with ovarian cancer who had a staging procedure for early stages, and primary or interval debulking surgery for advanced stages by laparoscopy or open surgery. Only comparative studies between laparoscopy and laparotomy were included in the present review. References of the included papers were further searched to identify other potentially relevant studies. Exclusion criteria included duplicate publications, nonEnglish language literature, abstracts, letters, editorials and reviews not reporting original data, studies with less than 10 patients and studies including patients treated for recurrent disease or fertility-sparing surgery only. Studies including patients treated by robot-assisted laparoscopy were also included if it was possible to distinguish their data from patients treated by laparoscopy. Additionally, we excluded studies that included borderline tumors when it was not possible to discern data related to women with invasive cancer. The Methodological Index for Nonrandomized Studies (MINORS) [15] was used to conduct quality evaluation for the studies.

In each report, we sought to extract oncologic outcomes, surgical details and baseline demographic data. Oncologic outcomes included five and three-year mortality rate and five and three-year recurrence rate. Surgery-related details included the following surgical related outcomes: mean operative time, mean blood loss, intraoperative complication rate, length of hospital stay and postoperative complications rate (all grades first, then we considered separately complications of grade ≥ 3 according to the Clavien-Dindo classification [16]). Demographic data were also searched: proportion of patients with adjuvant therapy, proportion of patients with neoadjuvant therapy and proportion of complete debulking surgery.

Studies were selected and data were extracted by two reviewers (FJ and MV), and any discrepancy between reviewers was resolved through discussion.

The main outcome measures were all-cause mortality within five and three years. Secondary outcomes measures were five and three-year recurrence, as well as the above-mentioned surgical outcomes. Data were presented as median values and ranges and were converted to mean values and standard deviations using the formula proposed by Wan et al. [17]. Survival data only available in Kaplan-Meier curves were extracted using the software Digitizeit (https://www.digitizeit.de). R 3.6.2 software was used to carry out the meta-analysis and the effect measures were presented with relative risk (RR)/mean difference (MD) and 95% confidence interval (CI). A random effect was used in the overall analysis due to the variability of the population included in each study. Subgroup analyses were conducted based on the initial characteristics of the study population: early stage only versus advanced ovarian cancer, and conducted using a mixed-effects model [18] in which subgroup effect sizes were pooled using a random-effects model, and subgroup differences were assessed using a fixed-effect model. Heterogeneity between studies was assessed using the Higgins I^2^ test, with levels of heterogeneity defined as not important (I^2^ = 0–40%), moderate (I^2^ = 30–60%), substantial (I^2^ = 50–90%), or considerable (I^2^ = 75–100%) [19]. If I^2^ ≥ 75%, data were considered to have considerable heterogeneity and could not be combined [19]. The χ^2^ test was used for the same purpose, with a statistical significance level of *p* < 0.05 indicating presence of statistical heterogeneity. A metaregression analysis was performed for survival outcomes to determine factors that had an influence on heterogeneity using the baseline demographics data mentioned above. The proportion of neoadjuvant therapy and the proportion of complete debulking were analyzed only in the advanced stage subgroup. Outcomes were given as the exponentiated slope coefficient and 95% CIs. Variables with *p* < 0.05 were regarded as significant influential factors on heterogeneity. Egger’s test [20] was utilized to evaluate publication bias. When there was a substantial publication bias, a Duval and Tweedie nonparametric trim and fill procedure was performed to assess the possible effects of the publication bias and to suggest the adjusted overall values. To conclude, a sensitivity analysis was conducted by detecting outliers in each meta-analysis. A study was defined as an outlier if the study’s confidence interval did not overlap with the confidence interval of the pooled effect. In that case, the study was removed from the analysis to examine the effect removal of the study had on the pooled effect.

## 3. Results

### 3.1. Literature Search

In this study, we enrolled 19 observational studies (Figure 1) [6,11,21,22,23,24,25,26,27,28,29,30,31,32,33,34,35,36,37]. In total, 7213 patients were included in this meta-analysis: 2285 (32%) in the minimally invasive surgery group and 4928 (68%) in the open surgery group. The design of each study, with the baseline demographic data, are provided in Table 1. The quality scores of the studies according to MINORS varied between 16 and 20 with a median value of 18.

### 3.2. Overall Mortality

A total of 10 studies reported all-cause five-year mortality (Figure 2). The meta-analysis revealed no significant difference for overall survival after MIS compared with OPS at five years (RR = 0.89, 95% CI 0.53–1.49, *p* = 0.62). No significant difference was observed between the early and advanced stage subgroups (*p* = 0.20). The statistical heterogeneity of the studies shows moderate heterogeneity in the early stage subgroup (I^2^ = 39%, 95% CI 0–74%, *p* = 0.13) and no heterogeneity in the advanced stage subgroup (I^2^ = 0%, 95% CI 0–80%, *p* = 0.59) with an overall heterogeneity of 50% (0–76%), *p* = 0.20. The funnel plot was symmetrical both according to visual and statistical testing (Egger test *p* = 0.97), arguing against small-study effects or publication bias. Metaregression found no significant result in subgroup analysis and overall (Table 2). In the advanced stage subgroup, the metaregression was not realized for adjuvant therapy due to the presence of only two studies. No outlier was detected in the sensitivity analysis.

Three-year mortality was reported in 14 studies (Figure 2). Three-year mortality for MIS compared with OPS was not significantly improved (RR = 0.95, 95% CI 0.80–1.12, *p* = 0.52). No significant difference was observed between the early and advanced stage subgroups (*p* = 0.30). The statistical heterogeneity of the studies showed no heterogeneity in the early stage subgroup (I^2^ = 0%, 95% CI 0–0%, *p* = 0.98), and moderate heterogeneity in the advanced stage subgroup (I^2^ = 39%, 95% CI 0–76%, *p* = 0.14), with an overall heterogeneity of 13% (0–52%), *p* = 0.30. The funnel plot appeared asymmetrical with a statistical testing significant (Egger test *p* = 0.01), indicating some level of small-study effects or publication bias. Relative risk was corrected using the trim and fill procedure and revealed an adjusted value of 0.97 (0.80–1.18) (Figure 3). As a result of metaregression, three-year survival was not associated with the proportion of neoadjuvant and adjuvant therapy (Table 2). From the advanced stage subgroup, the RR for three-year mortality was significantly reduced with a higher proportion of complete resection (*p* = 0.01 with r2 = 100%) (Figure 4). The sensitivity analysis found no outlier.

### 3.3. Secondary Outcomes

A total of nine studies reported five-year recurrence (Appendix B). The pooled analysis found no significant difference when comparing MIS with OPS (RR = 0.97, 95% CI 0.72–1.31; *p* = 0.84). No significant difference was observed between the early and advanced stage subgroups (*p* = 0.23). The statistical heterogeneity of the studies showed not important heterogeneity in the early stage subgroup (I^2^ = 22%, 95% CI 0–66%, *p* = 0.27), and in the advanced stage subgroup (I^2^ = 51%, 95% CI 0–86%, *p* = 0.13), with an overall heterogeneity of 41% (0%; 73%), *p* = 0.23. The funnel plot was symmetrical both according to visual and statistical testing (Egger test *p* = 0.14), arguing against small-study effects or publication bias. Metaregression overall, and in subgroups, found no significant result (Table 2). Meta-regression in the advanced stage subgroup for adjuvant therapy was not realized due to the presence of only two studies. No outlier was detected in the sensitivity analysis.

Three-year recurrence was reported in 10 studies (Appendix B). The pooled analysis showed no significant difference between MIS and OPS for three-year recurrence (RR = 0.94, 95% CI 0.73–1.22, *p* = 0.60). No significant difference was observed between the early and advanced subgroups (*p* = 0.23). The statistical heterogeneity of the studies showed no important heterogeneity in the early stage subgroup (I^2^ = 4%, 95% CI 0–76%, *p* = 0.39) and in the advanced stage subgroup (I^2^ = 20%, 95% CI 0–88%, *p* = 0.29, with an overall heterogeneity of 16% (0%; 57%), *p* = 0.23. The funnel plot was symmetrical both according to visual and statistical testing (Egger test *p* = 0.21), arguing against small-study effects or publication bias. Metaregression found no significant result (Table 2). No outlier was detected in the sensitivity analysis.

For estimated blood loss, 12 studies were initially included in the analysis (Appendix C). The statistical heterogeneity of the studies showed considerable heterogeneity in the early stage subgroup (I^2^ = 76%, 95% CI 51–89%, *p* < 0.01) and in the advanced stage subgroup (I^2^ = 95%, 95% CI 92–97%, *p* < 0.01), with an overall heterogeneity of 90% (85%; 94%), *p* < 0.001. A significant difference was observed between the early and advanced subgroups (*p* = 0.02). The sensitivity analysis found one outlier in the early stage subgroup and three outliers in the advanced stage subgroup. These studies were removed from the analysis. The statistical heterogeneity of the remaining eight studies showed moderate heterogeneity in the early stage subgroup (I^2^ = 33%, 95% CI 0–73%, *p* = 0.19), and not important heterogeneity in the advanced stage subgroup (I^2^ = 14%, *p* = 0.28). A significant difference was still observed between the two subgroups (*p* < 0.001) with, in the early stage subgroup, an MD for estimated blood loss at −187.99 (−239.91; −134.07) and in advanced stage subgroup at −1167.84 (−1673.25; −662.42). The pooled analysis could not be realized due to high heterogeneity (I^2^ = 73%, 95% CI 45–97%, *p* < 0.01).

A total of 14 studies were initially included in the meta-analysis for operating time (Appendix C). The statistical heterogeneity of the studies showed considerable heterogeneity in the early stage subgroup (I^2^ = 89%, 95% CI 80–94%, *p* < 0.01), and moderate heterogeneity in the advanced stage subgroup (I^2^ = 33%, 95% CI 0–71%, *p* = 0.18) with an overall heterogeneity at 80% (67%; 88%), *p* = 0.37. No significant difference was observed between the early and advanced stage subgroups (*p* = 0.37). One outlier was removed from each subgroup in the sensitivity analysis. The pooled analysis of the remaining 12 studies showed no significant difference for operative time between MIS and OPS (MD = 8.89, 95% CI −0.01 to 17.8, *p* = 0.05; I^2^ = 43%, 95% CI 0%–71%, *p* = 0.33). The funnel plot was symmetrical both according to visual and statistical testing (Egger test *p* = 0.64), arguing against small-study effects or publication bias.

Mean hospital stay was reported in 16 studies (Appendix C). The statistical heterogeneity of the studies showed considerable heterogeneity in the early stage subgroup (I^2^ = 95%, 95% CI 93–97%, *p* < 0.01), and in the advanced stage subgroup (I^2^ = 93%, 95% CI 88–96%, *p* < 0.01) with an overall heterogeneity at 94% (92%; 96%), *p* = 0.10. No significant difference was observed between the early and advanced stage subgroups (*p* = 0.10). The sensitivity analysis removed seven outliers. The pooled analysis of the remaining nine studies showed a significant reduction of hospital stay when patients were treated with MIS (MD = −2.59, 95% CI −3.22 to −1.97, *p* < 0.01; I^2^ = 57%, 95% CI 9–80%, *p* = 0.64). The funnel plot was symmetrical both according to visual and statistical testing (Egger test *p* = 0.71), arguing against small-study effects or publication bias.

As for intraoperative complication rate, 11 studies were included in the meta-analysis (Appendix C). The pooled analysis revealed no significant difference between MIS and OPS (RR = 0.77, 95% CI 0.51–1.16, *p* = 0.18). No significant difference was observed between the early and advanced stage subgroups (*p* = 0.88). The statistical heterogeneity of the studies showed no heterogeneity in the early stage subgroup (I^2^ = 0%, 95% CI 0–0%, *p* = 0.95) and in the advanced stage subgroup (I^2^ = 0%, 95% CI 0–66%, *p* = 0.66) with an overall heterogeneity at 0% (0%; 0%), *p* = 0.88. The funnel plot appeared asymmetrical according to visual observation, even if statistical testing was not significant (Egger test *p* = 0.06), indicating the possibility of small-study effects or publication bias. The relative risk was corrected using the trim and fill procedure and revealed an adjusted value of 0.97 (0.61–1.52) (Appendix D). No outlier was detected in the sensitivity analysis.

Finally, postoperative all grade complication rate was analyzed in nine studies (Appendix C). The meta-analysis showed an almost significant reduction of postoperative complication in favor of MIS (RR = 0.65, 95% CI 0.40–1.05, *p* = 0.07). No significant difference was observed between the early and advanced stage subgroups (*p* = 0.31). The statistical heterogeneity of the studies showed no heterogeneity in the early stage subgroup (I^2^ = 4%, 95% CI 0–85%, *p* = 0.37) and substantial heterogeneity in the advanced stage subgroup (I^2^ = 57%, 95% CI 0–69%, *p* = 0.05, with an overall heterogeneity at 33% (0%; 69%), *p* = 0.31. The funnel plot was symmetrical both according to visual and statistical testing (Egger test *p* = 0.22), arguing against small-study effects or publication bias. No outlier was detected in the sensitivity analysis. As for postoperative complication rate grade ≥3, only five studies recorded it. The low number of studies can be explained by the high number of different definitions of perioperative complications. The pooled analysis showed, again, no significant difference between MIS and OPS (RR = 0.67, 95% CI 0.25–1.80, *p* = 0.32). No significant difference was observed between the early and advanced stage subgroups (*p* = 0.76). The statistical heterogeneity of the studies showed no heterogeneity in the early stage subgroup (I^2^ = 0%, *p* = 0.70) and not important heterogeneity in the advanced stage subgroup (I^2^ = 21%, 95% CI 0–92%, *p* = 0.76), with an overall heterogeneity at 0% (0%; 69%), *p* = 0.09. The funnel plot was symmetrical both according to visual and statistical testing (Egger test *p* = 0.56), arguing against small-study effects or publication bias. No outlier was detected in the sensitivity analysis.

## 4. Discussion

This report is, to our knowledge, the only quantitative meta-analysis to date to compare the five-year survival between MIS and OPS in ovarian cancer. We found that patients diagnosed with ovarian cancer undergoing MIS presented similar five-year (RR = 0.89, 95% CI 0.53–1.49, *p* = 0.62) and three-year mortality (RR = 0.95, 95% CI 0.80–1.12, *p* = 0.52). When five-year and three-year recurrences were analyzed, no statistically significant differences were observed. Analyses in subgroups show no significant difference for survival outcomes, suggesting an oncological safety of MIS in both early and advanced stages. Metaregression found no impact of the proportion of neoadjuvant and adjuvant chemotherapy on survival outcomes. Whether the surgery was primary or interval debulking surgery in advanced ovarian cancer, it did not influence the comparative results on mortality or recurrence. Metaregression on three-year mortality found, however, a significant influence of the proportion of complete resection in advanced ovarian cancer. When the proportion of complete resection is higher, the three-year mortality is lower with MIS compared to OPS. This result reinforces the idea that, if patients are correctly selected, minimally invasive surgery can be a very effective treatment. Unfortunately, these selection criteria have yet to be precisely defined, which is currently one of the main barriers to the acceptance of laparoscopic management of ovarian cancer. 

In all the studies included, no specific criteria were used to select patients for laparoscopy, and there was great heterogeneity in the way the groups were set up. Minimally invasive surgery should be reserved only in centers that might guarantee the possibility of complete cytoreduction when judged to be feasible. The analyses of perioperative outcomes showed a decrease in morbidity with a reduction in the length of hospital stay with MIS. Although estimated blood loss could not be aggregated overall, the subgroups analyses showed a significant reduction of blood loss with MIS. No significant differences were observed for operative time, intraoperative complications and postoperative complications, even if the all grade complications analyses were at the borderline of significance (RR = 0.65, 95% CI 0.40–1.05, *p* = 0.07).

These findings must be interpreted in the context of several important caveats. First, no randomized clinical trials comparing MIS and OPS in ovarian cancer were identified, and so our meta-analysis included only observational studies, which were mostly retrospective. This can lead to the presence of potential bias. Therefore, conclusions should only be regarded as hypothetical conclusions and not as absolute truth. However, few publication biases were identified in our study. Second, in many of the studies included, the MIS and OPS cohorts differed with respect to prognostically important variables. The metaregression was adjusted for important prognostic factors, yet residual confounding cannot be excluded. Patients in the laparoscopy group often had fewer complex procedures, which could be an indirect expression of lower burden of disease. Unfortunately, we were not able to extract patient-level data regarding these variables, especially the surgical complexity. Biases in the individual studies might have affected the results of the meta-analysis. Should data of this type become available, a more robust analysis based on these variables could be performed.

## 5. Conclusions

Although the available studies are retrospective, and mostly carry a high risk for bias and confounding, an overwhelming consistency of the evidence suggests the likely effectiveness of MIS in selected cases of ovarian cancer, even in advanced stages. In light of the existing evidence, we further recommend that additional retrospective cohort trials will not contribute additional useful data. In order to validate this conclusion on the oncological safety of MIS, conducting a feasibility study followed by a randomized clinical trial should be a priority.

## Figures and Tables

**Figure 1 jcm-09-02507-f001:**
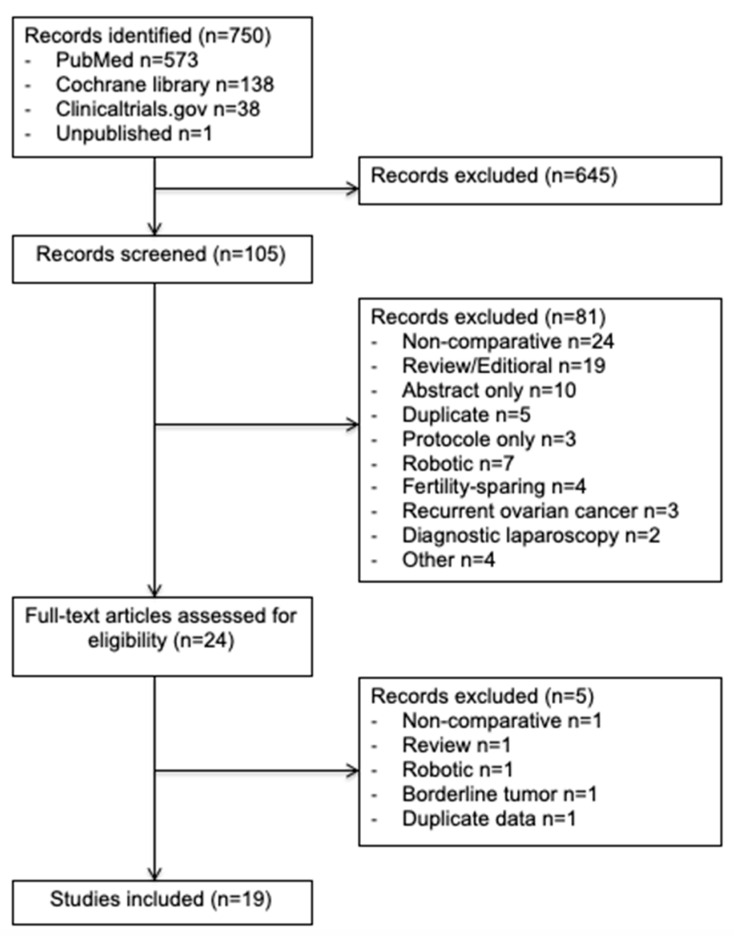
Preferred Reporting Items for Systematic Reviews and Meta-Analyses (PRISMA) Flow Diagram.

**Figure 2 jcm-09-02507-f002:**
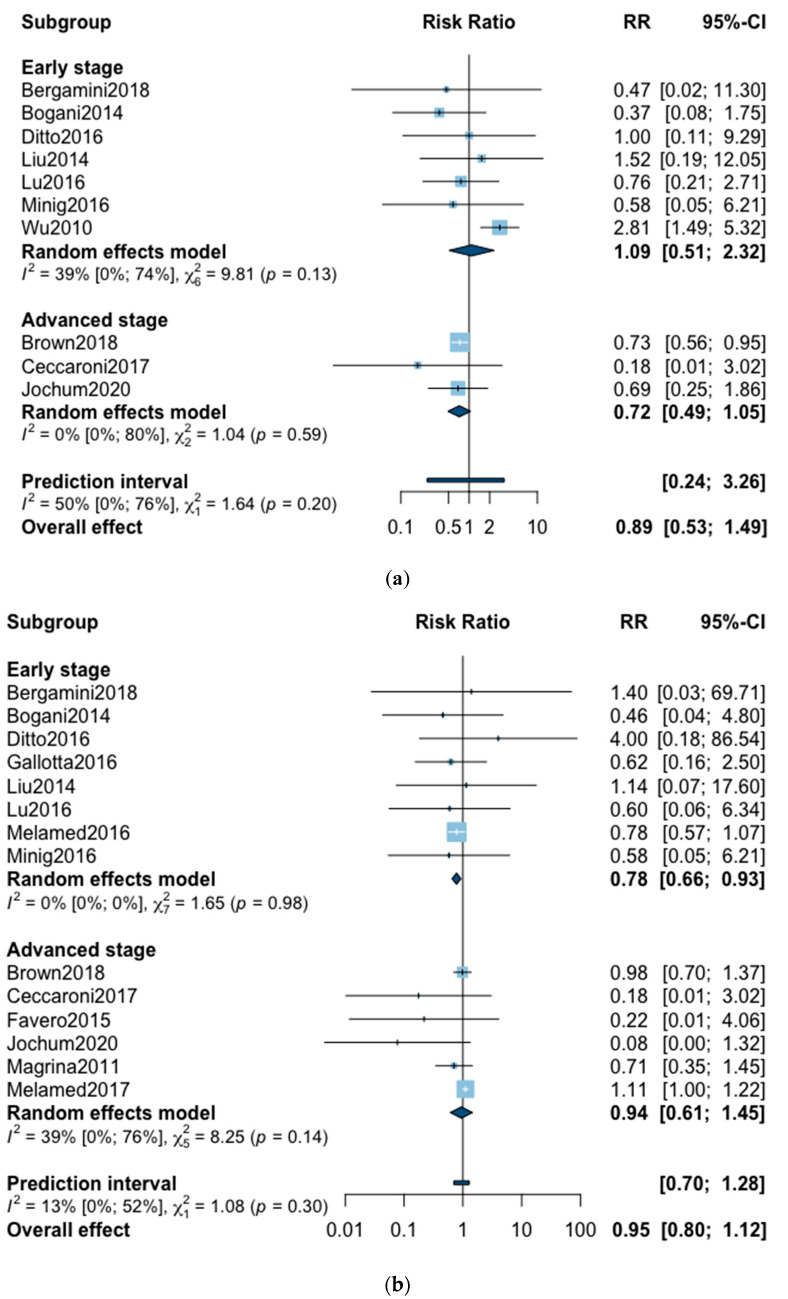
Forrest plot for relative risk (RR) for five-year mortality (**a**) and three-year mortality (**b**).

**Figure 3 jcm-09-02507-f003:**
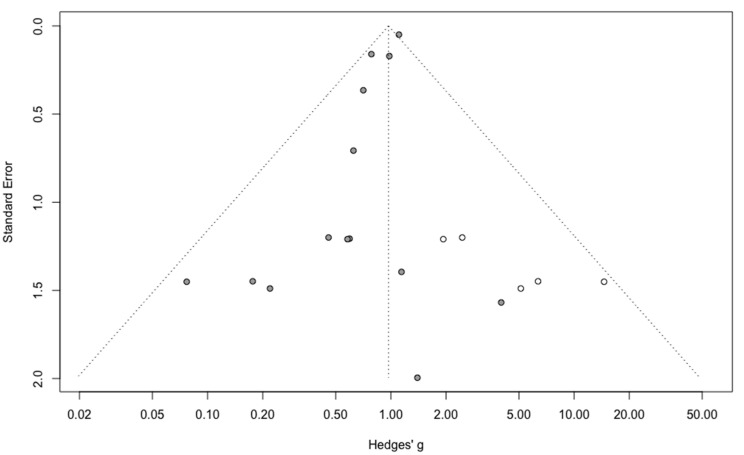
Corrected funnel plot for three-year mortality outcome.

**Figure 4 jcm-09-02507-f004:**
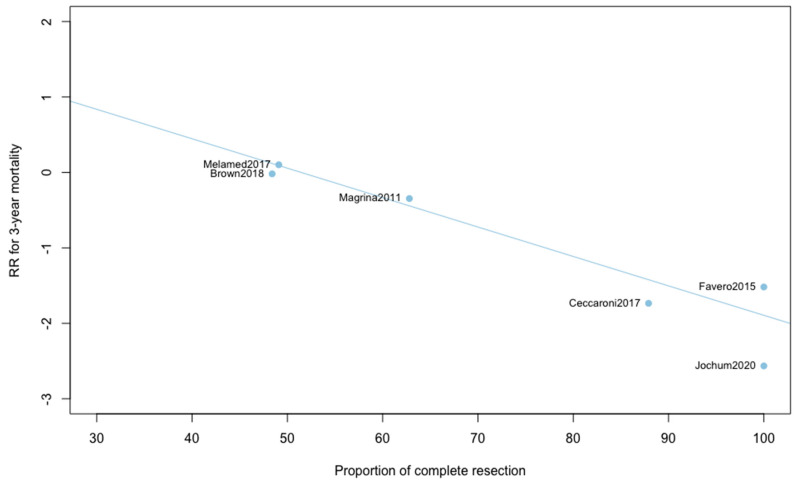
Metaregression of RR for 3-year mortality by proportion of complete resection. y = 2.01 − 0.04x.

**Table 1 jcm-09-02507-t001:** Study design and demographic data.

Study	Period	Study Design	Total MINORS *	Location	Histological Type	Number of Patients	Follow-Up, Months	Adjuvant Therapy, *n* (%)	Neoadjuvant Therapy, *n* (%)	Complete Resection, *n* (%)
Early ovarian cancer										
Bergamini et al., 2018	1965–2017	Retrospective	18	Multicenter Italy	Granulosa cells	MIS = 93	81 (10–450)	25 (11%)	-	-
						OS = 130				
Bogani et al., 2014	2003–2010	Retrospective	18	Monocentric Italy	Epithelial	MIS = 35	64 (37–106)	56 (84%)	-	-
						OS = 32	100 (61–287)			
Ditto et al., 2016	2005–2015	Retrospective and prospective	19	Monocentric Italy	Epithelial	MIS = 50	49.5 (64)	59 (59%)	-	-
						OS = 50	52.6 (31.7)			
Gallotta et al., 2016	2000–2013	Retrospective	17	Monocentric Italy	Epithelial	MIS = 60	38 (24–48)	126 (70%)	-	-
						OS = 120				
Koo et al., 2014	2006–2012	Retrospective	18	Multicenter Korea	All types	MIS = 24	31.7 (20.7)	69 (90%)	-	-
						OS = 53	31.1 (19.1)			
Lee et al., 2011	2005–2010	Retrospective	18	Monocentric Korea	All types	MIS = 26	12 (1–42)	82 (73%)	-	-
						OS = 87	25 (1–74)			
Liu et al., 2014	2002–2012	Retrospective	16	Monocentric China	All types	MIS = 35	36 to 84	66 (88%)	-	-
						OS = 40				
Lu et al., 2016	2002–2014	Retrospective	19	Monocentric China	Epithelial	MIS = 42	82 (16–152)	-	-	-
						OS = 50	82 (16–152)			
Melamed et al., 2016	2010–2012	Retrospective	20	Multicenter USA	Epithelial	MIS = 1096	28.7 (20.4–38.9)	1230 (56%)	-	-
						OS = 1096	29.3 (20.6–39.3)			
Minig et al., 2016	2006–2014	Retrospective	17	Multicenter Spain and Argentina	Epithelial	MIS = 50	25.9 (11.2–38.5)	66 (61%)	-	-
						OS = 58	34.3 (32.8–49)			
Wu et al., 2009	1984–2006	Retrospective	18	Multicenter Taiwan	Epithelial	MIS = 34	48.5 (3–174.5)	152 (78%)	-	-
						OS = 174	67 (2–276)			
Advanced ovarian cancer										
Alletti et al., 2016	2010–2014	Retrospective	17	Monocentric Italy	High grade serous	MIS = 30	28	95 (100%)	95 (100%)	91 (96%)
						OS = 65				
Brown et al., 2018	2006–2017	Retrospective	19	Monocentric USA	Epithelial	MIS = 53	-	-	157 (100%)	76 (48%)
						OS = 104				
Ceccaroni et al., 2017	2007–2015	Prospective	19	Monocentric Italy	All types	MIS = 21	47.3 (12–72)	66 (100%)	0	58 (88%)
						OS = 45	52.3 (5–117)			
Favero et al., 2015	2011–2014	Prospective	20	Monocentric Brazil	High grade serous	MIS = 10	20 (12–26)	21 (100%)	21 (100%)	21 (100%)
						OS = 11	36 (24–48)			
Jochum et al., 2020	2010–2018	Retrospective	19	Monocentric France	Epithelial	MIS = 41	31.0 (16.0–52.0)	67 (82%)	45 (55%)	82 (100%)
						OS = 41	32.0 (23.0–61.0)			
Magrina et al., 2011	2002–2008	Retrospective	17	Monocentric USA	Epithelial	MIS = 27	52.8 (2.4–110.4)	106 (73%)	36 (25%)	92 (63%)
						OS = 119	34.8 (0–128.4)			
Melamed et al., 2017	2010–2012	Retrospective	19	Multicenter USA	Epithelial	MIS = 540	32.0	-	3161 (100%)	919 (49%)
						OS = 2621				
Tozzi et al., 2016	2008–2016	Prospective	19	Multicenter Italy and United Kingdom	All type	MIS = 18	-	-	50 (100%)	49 (98%)
						OS = 32	-			

* Methodological Index for Nonrandomized Studies.

**Table 2 jcm-09-02507-t002:** Metaregression analysis.

Metaregression	k	Exponentiated Slope Coefficient (95% CI)	*p* Value
Five-year mortality			
Metaregression by adjuvant therapy, %			
Early stage	6	0.01 (−0.04 to 0.07)	0.55
Overall	8	−0.00 (−0.05 to 0.05)	0.99
Metaregression by neoadjuvant therapy, %			
Advanced stage	3	0.01 (−0.08 to 0.09)	0.51
Metaregression by complete resection, %			
Advanced stage	3	−0.00 (−0.12 to 0.11)	0.77
Three-year mortality			
Metaregression by adjuvant therapy, %			
Early stage	7	−0.01 (−0.04 to 0.02)	0.53
Advanced stage	4	−0.05 (−0.20 to 0.09)	0.24
Overall	11	−0.02 (−0.04 to 0.00)	0.06
Metaregression by neoadjuvant therapy, %			
Advanced stage	5	0.01 (−0.01 to 0.02)	0.21
Metaregression by complete resection, %			
Advanced stage	6	−0.04 (−0.06 to −0.02)	0.01
Five-year recurrence			
Metaregression by adjuvant therapy, %			
Early stage	6	0.00 (−0.02 to 0.03)	0.69
Overall	8	0.00 (−0.01 to 0.01)	0.78
Metaregression by neoadjuvant therapy, %			
Advanced stage	3	−0.01 (−0.04 to 0.03)	0.25
Meta-regression by complete resection, %			
Advanced stage	3	0.01 (−0.02 to 0.03)	0.14
Three-year recurrence			
Metaregression by adjuvant therapy, %			
Early stage	6	−0.01 (−0.03 to 0.02)	0.58
Advanced stage	3	0.03 (−0.04 to 0.10)	0.12
Overall	9	−0.00 (−0.02 to 0.01)	0.66
Metaregression by neoadjuvant therapy, %			
Advanced stage	4	−0.00 (−0.02 to 0.02)	0.73
Metaregression by complete resection, %			
Advanced stage	4	0.01 (−0.03 to 0.04)	0.55

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
