# Peer review of "Three and Five-Year Mortality in Ovarian Cancer after Minimally Invasive Compared to Open Surgery: A Systematic Review and Meta-Analysis"

_jcm, 2020, doi:10.3390/jcm9082507_

Round 1
Reviewer 1 Report
RE: “Three and five-year mortality in ovarian cancer after minimally invasive compared to open surgery: a systematic review and meta-analysis
The research paper written by Jochum and colleagues, 2020 under consideration for publication has presented a meta-analysis of available observational studies assessing the role of MIS vs open surgery in ovarian cancer. The meta-analysis included 15 studies and 7213 individuals, with both early and advanced stage.
This is a thoughtful and comprehensive meta-analysis addressing an important topic in gynecologic oncology surgery. Please find immediately below, several items for the authors to address.
Areas for further clarifications in the manuscript:
-Consider using a different acronym for open surgery. The term OS is widely used for overall survival and it can be confusing.
- English language editing is required
Abstract:
-Consider including a short introduction addressing the role of MIS in invasive ovarian cancer.
-Consider including the number of patients identified from the 19 studies in the abstract.
- “Studies were selected and data were extracted by 2 reviewers.” This is standard for meta-analysis, maybe not necessary to include in the abstract, only in the body of the text.
- “We finally enrolled 19 observational studies”. I think that it should read included instead of enrolled, as this is not a trial were patients are enrolled.
- Efficacy: “We found that patients diagnosed with ovarian cancer undergoing MIS presented similar 5-year (RR=0.89, 95% CI 0.53– 1.49, p=0.62) and 3-year mortality (RR=0.95, 95% CI 0.80–1.12, p=0.52). When 5 and 3-year recurrence were analyzed, no statistically significant difference was observed.” Consider using no statistically significant difference for both mortality and recurrence.
-“Analyses in subgroups show no significant difference for survival outcomes, suggesting an oncological safety of MIS in both early and advanced stages.” Consider rephrasing: “Analysis in early and advance stage subgroups… for survival outcomes” for clarity, instead of listing the subgroups at the end of the sentence.
-“Although the available studies are retrospective and mostly carry a high risk for bias and confounding, an overwhelming consistency of the evidence supports the efficiency of MIS in selected cases of ovarian cancer, even in advanced stages”, consider rephrasing this. In my opinion there is not enough data yet to support MIS in selected cases, and you don’t have “efficiency” data. Instead, I would say that it supports the development of future randomized interventional studies.
Introduction
-Line 28 “oncological safety in the longer term”. Should it read outcomes instead of safety?
-Line 31 “Meta-analyses of ovarian cancer have recently been published…”. Consider summarizing their findings and not only what is missing.
-Consider including a short summary of upfront vs interval debulking, and the importance of no residual disease, as this is described later on in the results.
Experimental section
-Was the meta-analysis registered in PROSPERO? Please include the registration number in the manuscript.
Results:
-In several areas in the manuscript you mention between the two subgroups (for example page 7, 112; page 12, 147), and it is not very clear whether you compare early and advanced stage subgroups. I think it would be helpful if you can clarify that through the text.
Conclusions:
-“Randomized clinical trials have not proved feasible. In order to validate this conclusion on the oncological safety of MIS, a high-quality, multicenter, audited, prospective registry of patients treated by MIS for ovarian cancer is a valuable alternative and should be a priority.” In my opinion next steps to assess MIS should be a feasibility study followed by a randomized trial. I do not think that the scientific community will change the standard of care practice with a prospective registry that will still be biased. Several randomized trials have been performed assessing surgical techniques in gynecological cancers, including the LEAP trial assessing MIS vs open surgery in cervical cancer.
Author Response
Please find below our point-by-point answers for the reviewer 1.
-Consider using a different acronym for open surgery. The term OS is widely used for overall survival and it can be confusing.
We agree with the reviewer. It may be confusing. We changed the acronym for open surgery by OPS.
Abstract:
-Consider including a short introduction addressing the role of MIS in invasive ovarian cancer.
As requested, we added a short introduction addressing the role of MIS in invasive ovarian cancer :
Line 5 "As regards ovarian cancer, the use of minimally invasive surgery has steadily increased over the years. Reluctances persist, however, about its oncological outcomes."
-Consider including the number of patients identified from the 19 studies in the abstract.
We have added the number of patients included line 11: "We finally included 19 observational studies with a total of 7213 patients"
-“Studies were selected and data were extracted by 2 reviewers.” This is standard for meta-analysis, maybe not necessary to include in the abstract, only in the body of the text.
This sentence was removed from the abstract.
-“We finally enrolled 19 observational studies”. I think that it should read included instead of enrolled, as this is not a trial were patients are enrolled
This sentence was modified:
Line 11 "We finally included 19 observational studies with a total of 7213 patients"
-Efficacy: “We found that patients diagnosed with ovarian cancer undergoing MIS presented similar 5-year (RR=0.89, 95% CI 0.53– 1.49, p=0.62) and 3-year mortality (RR=0.95, 95% CI 0.80–1.12, p=0.52). When 5 and 3-year recurrence were analyzed, no statistically significant difference was observed.” Consider using no statistically significant difference for both mortality and recurrence.
We modified this sentence :
"We found no statistically significant difference for 5-year (RR=0.89, 95% CI 0.53– 1.49, p=0.62) and 3-year mortality (RR=0.95, 95% CI 0.80–1.12, p=0.52) between the patients undergoing MIS and those operated by OPS. When 5 and 3-year recurrence were analyzed, no statistically significant difference was also observed."
-“Analyses in subgroups show no significant difference for survival outcomes, suggesting an oncological safety of MIS in both early and advanced stages.” Consider rephrasing: “Analysis in early and advance stage subgroups… for survival outcomes” for clarity, instead of listing the subgroups at the end of the sentence.
The sentence was modified:
"Analysis in early and advanced stages subgroups show no significant difference for survival outcomes, suggesting an oncological safety of MIS in all stages."
-“Although the available studies are retrospective and mostly carry a high risk for bias and confounding, an overwhelming consistency of the evidence supports the efficiency of MIS in selected cases of ovarian cancer, even in advanced stages”, consider rephrasing this. In my opinion there is not enough data yet to support MIS in selected cases, and you don’t have “efficiency” data. Instead, I would say that it supports the development of future randomized interventional studies.
A modification of this sentence was made:
"Although the available studies are retrospective and mostly carry a high risk for bias and confounding, an overwhelming consistency of the evidence suggests the likely effectiveness of MIS in selected cases of ovarian cancer, even in advanced stages. To validate the use of MIS, the development of future randomized interventional studies should be a priority."
Introduction
-Line 28 “oncological safety in the longer term”. Should it read outcomes instead of safety?
The sentence was modified:
Line 35 "Reluctances persist, however, about its oncological outcomes in the longer term."
-Line 31 “Meta-analyses of ovarian cancer have recently been published…”. Consider summarizing their findings and not only what is missing.
This sentence was modified in order to summarize quickly the findings of the meta-analyses:
Line 38 "Meta-analyses of ovarian cancer have recently been published[5–12], showing similar operative and clinical outcomes between patients treated by MIS and those operated by OPS. However, none of them compared 3 or 5-year survival, or performed an overall analysis (early and advanced stages)."
-Consider including a short summary of upfront vs interval debulking, and the importance of no residual disease, as this is described later on in the results.
The introduction was completed by a short summary of upfront vs interval debunking, and the importance of no residual disease:
Line 27 "Surgery, together with chemotherapy, are the pillars of the management of ovarian cancer. For early stages, the main objective is to establish the stage of the disease in the purpose to confirm the indication of adjuvant chemotherapy. For advanced tumors, the mainstay of the curative treatment is radical cytoreduction without any residual disease, followed by chemotherapy [2]. Whenever this finality is unachievable at upfront surgery, neo-adjuvant chemotherapy and interval debulking surgery was accepted as a valid alternative."
Experimental section
-Was the meta-analysis registered in PROSPERO? Please include the registration number in the manuscript.
This meta-analysis was registered in PROSPERO. The registration number is CRD42020183284. It has been had in the manuscript:
Line 90 "It has been registered in PROSPERO (CRD42020183284)."
Results:
-In several areas in the manuscript you mention between the two subgroups (for example page 7, 112; page 12, 147), and it is not very clear whether you compare early and advanced stage subgroups. I think it would be helpful if you can clarify that through the text.
Clarifications at different points in the manuscript have been made. For example: Line 173 "No significant difference was observed between the early and advanced stage subgroups (p=0.20)."
Conclusions:
-“Randomized clinical trials have not proved feasible. In order to validate this conclusion on the oncological safety of MIS, a high-quality, multicenter, audited, prospective registry of patients treated by MIS for ovarian cancer is a valuable alternative and should be a priority.” In my opinion next steps to assess MIS should be a feasibility study followed by a randomized trial. I do not think that the scientific community will change the standard of care practice with a prospective registry that will still be biased. Several randomized trials have been performed assessing surgical techniques in gynecological cancers, including the LEAP trial assessing MIS vs open surgery in cervical cancer.
We agree with the reviewer. A modification of the conclusion has been made:
"Although the available studies are retrospective and mostly carry a high risk for bias and confounding, an overwhelming consistency of the evidence suggests the likely effectiveness of MIS in selected cases of ovarian cancer, even in advanced stages. In light of the existing evidence, we further recommend that additional retrospective cohort trials will not contribute additional useful data. In order to validate this conclusion on the oncological safety of MIS, conducting a feasibility study followed by a randomized clinical trial should be a priority."
Reviewer 2 Report
Introduction
This manuscript presents a detailed retrospective meta-analysis (19 global studies) comparing both the 3 and 5-year survival, along with respective perioperative outcomes of ovarian cancer patients operated by minimally invasive surgery compared to conventional open surgery, overall no significant differences were observed across all analysis suggesting an oncological safety of minimally invasive surgery across all stages of the disease. Overall, this was a very detailed and well-written analysis that will validates the need for large prospective trial study.
Minor revisions
Line 32 – performed instead of perform
Figure 1,2, 4 are out of focus in PDF
The meta-regression adjusted for important prognostic factors was great. However, did the authors look into whether analysis broken down into ovarian histological subtypes (high and low grade serous, mucinous, clear cell and endo) showed comparative results on mortality or recurrence? Or was this patient level data difficult to extract?
Show subtypes distribution in supplementary or in methods
Author Response
Please find below our point-by-point answers for the reviewer 2.
Line 32 – performed instead of perform
We modified this sentence and replaced perform by performed.
Figure 1,2, 4 are out of focus in PDF
You are right. They do seem a bit blurry due to the uploading on Word. It does not affect however the figures including in the zip folder send to the journal for publication.
The meta-regression adjusted for important prognostic factors was great. However, did the authors look into whether analysis broken down into ovarian histological subtypes (high and low grade serous, mucinous, clear cell and endo) showed comparative results on mortality or recurrence? Or was this patient level data difficult to extract?
Thank you for your comment. At the time of writing the protocol, we did indeed think of analyzing the effect of the histological subtype on recurrence and mortality. However, one of the pitfalls of meta-regression is the false positive conclusions due to multiple analyses. In view of the number of studies included in our meta-analysis and in order to limit these false positive conclusions, we decided to limit the number of variables analyzed in the meta-regression. We chose to keep neoadjuvant therapy, adjuvant therapy and complete resection. Moreover, this patient level data is indeed difficult to extract and not available in many studies.
Show subtypes distribution in supplementary or in methods
As detailed above, subtypes distribution are sadly not available in many studies, or are displayed in various ways. We have added a column in Table 1 to provide a big picture perspective of the ovarian histological types in each study.